# Trends in Ultra-Processed Food Purchases from 1984 to 2016 in Mexican Households

**DOI:** 10.3390/nu11010045

**Published:** 2018-12-26

**Authors:** Joaquín Alejandro Marrón-Ponce, Lizbeth Tolentino-Mayo, Mauricio Hernández-F, Carolina Batis

**Affiliations:** 1Center for Nutrition and Health Research, National Institute of Public Health, Cuernavaca 62100, Morelos, Mexico; joaquin.marron@hotmail.com (J.A.M.-P.); mltolentino@insp.mx (L.T.-M.); mhf1980@gmail.com (M.H.-F.); 2CONACYT-Center for Nutrition and Health Research, National Institute of Public Health, Cuernavaca 62100, Morelos, Mexico

**Keywords:** trends, ultra-processed foods, purchases, Mexican households

## Abstract

Global trade agreements have shaped the food system in ways that alter the availability, accessibility, affordability, and desirability of ready-to-eat foods. We assessed the time trends of ultra-processed foods purchases in Mexican households from 1984 to 2016. Cross-sectional data from 15 rounds of the National Income and Expenditure Survey (1984, 1989, 1992, 1994, 1996, 1998, 2000, 2002, 2004, 2006, 2008, 2010, 2012, 2014 and 2016) were analyzed. Food and beverage purchases collected in a daily record instrument (over seven days) were classified according to their degree of processing according to the NOVA food framework: (1) Unprocessed or minimally processed foods; (2) processed culinary ingredients; (3) processed foods; and (4) ultra-processed foods. From 1984 to 2016, the total daily energy purchased decreased from 2428.8 to 1875.4 kcal/Adult Equivalent/day, there was a decrease of unprocessed or minimally processed foods (from 69.8% to 61.4% kcal) and processed culinary ingredients (from 14.0% to 9.0% kcal), and an increase of processed foods (from 5.7% to 6.5% kcal) and ultra-processed foods (from 10.5% to 23.1% kcal). Given that ultra-processed foods purchases have doubled in the last three decades and unprocessed or minimally processed foods purchased have gradually declined, future strategies should promote the consumption of unprocessed or minimally processed foods, and discourage ultra-processed foods availability and accessibility in Mexico.

## 1. Introduction

Since the 1980s, the food industry has dominated the food system with marketing strategies encouraging consumers to eat more ready-to-eat products in their diet [1]. Moreover, in the 1990s, international and global trade agreements intensified its role and influence across the globe [2]. Consequently, these changes shaped the food system in ways that alter the availability, accessibility, affordability, and desirability of ready-to-eat foods. Hence, these products are now an important energy source in the dietary patterns of the population [3,4,5,6,7,8]. To understand these changes and their consequences in dietary quality and health, the NOVA food framework is now recognized as a tool to classify foods according to their degree of processing (unprocessed or minimally processed foods; processed culinary ingredients; processed foods; and ultra-processed foods) [9].

Nationally representative dietary studies conducted with the NOVA food framework have indicated that a high dietary energy contribution of ultra-processed foods is associated with a lower dietary quality, which is characterized by a higher intake of free sugar and saturated fat, and by a lower intake of protein and dietary fiber [3,4,7,10]. Moreover, the consumption of these products has been associated with obesity in adults [11], metabolic syndrome in adolescents [12], and higher total cholesterol and LDL cholesterol in children [13].

In Mexico, ultra-processed food retail sales grew 29.2% (from 164.3 to 212.2 kg per capita/year) between 2000 and 2013 [14]. As a consequence of this increase, Mexico was placed as the first consumer of ultra-processed foods in Latin America and the fourth across the world [14]. However, it is unknown how purchases of ultra-processed foods have changed over a longer period; and how did these changes relate to the purchases of the remaining food groups. Therefore, the objective of this study was to assess time trends of ultra-processed foods purchases in Mexican households from 1984 to 2016. Studying the evolution of ultra-processed foods purchases over a 32-year frame would help to understand the influence of different trades, policies, and economic events that have shaped the dietary patterns of the population. This type of information is worthy to report due to the epidemiological transition that Mexico is facing with high prevalence of obesity, diabetes, and hypertension [15].

## 2. Materials and Methods

### 2.1. Data Sources and Collection

This study used 15 rounds of the National Income and Expenditure Survey (Encuesta Nacional de Ingresos y Gastos de los Hogares (ENIGH)) conducted by the National Institute of Statistics and Geography of Mexico (Instituto Nacional de Estadística y Geografía (INEGI)): 1984, 1989, 1992, 1994, 1996, 1998, 2000, 2002, 2004, 2006, 2008, 2010, 2012, 2014 and 2016. All ENIGH rounds were collected between August and November (except ENIGH 1994, which was collected between September and December). The ENIGH is a probabilistic based survey with a two-stage stratified clustered sampling design, representative at the national level, and urban and rural strata. It collects data on household expenditures in food and beverages, and sociodemographic characteristics such as household composition, income, head of household educational level, and place of residence (urban/rural). Detailed information about data collection is available elsewhere [16].

In each round, the number of households varied (from 4555 households in 1984 to 67,807 in 2016). In our analyses, we excluded households that did not report food and beverage expenditures (*n* = 25,241) and with incomplete sociodemographic information (*n* = 1427). We also excluded households that reported a total energy purchased ≥10,000 kcal/Adult Equivalent (AE)/day (~percentile 99.5) because this quantity was considered implausible for individual consumption (*n* = 1797). Thus, we analyzed the daily food and beverage purchases from a total sample of 277,509 Mexican households.

Daily food and beverage expenditure was collected for 1 week. On the first day, the enumerators trained the household member responsible for the food and beverage purchases to complete the daily record instrument for the next days. Besides these purchases (monetary expenses), he/she was also asked to record all those food and beverages obtained as a gift, through subsidized prices, or produced by the family (non-monetary expenses). Both types of expenses were deflected by the national consumer price index of November 2016 to enable comparability over time. In this instrument, the quantity (kg or lt) of both types of expenses were recorded for each food item reported. For beverages, lt were converted to kg by using their density. The list of food items in the instrument were: (1) Single foods (e.g., apple, lettuce, fish, egg, corn tortilla, sausages.) or (2) group of foods (e.g., white bread and baguette; smoked fish, dry fish, nugget and sardine; cider, wine, and vodka.).

### 2.2. Food Classification According to Processing

We classified the food and beverage items according to the NOVA framework into: (1) Unprocessed or minimally processed foods (*n* = 14 food subgroups); (2) processed culinary ingredients (*n* = 3 food subgroups); (3) processed foods (*n* = 4 food subgroups); and (4) ultra-processed foods (*n* = 10 food subgroups). When the food or beverage item was composed by a food group, we classified it into one NOVA group based on the classification of the majority of the food or beverages included in the item.

For each NOVA food group, we estimated the following indicators: (1) Percentage of consumers (we considered the household as a consumer if it reported any expenditure for that NOVA food group); (2) percentage of expenditure; (3) percentage of volume; and (4) percentage of energy purchased. We also estimated the total expenditure, volume, and energy purchased per day by AE. Energy content of purchased items was estimated using the food composition database compiled by the National Council for the Evaluation of Social Development Policy (Consejo Nacional de Evaluación de la Política de Desarrollo Social (CONEVAL)) and the AE was estimated by dividing the recommended dietary allowance (RDA) for energy of each household member (according to age and sex) by the energy RDA for an average adult (2550 kcal) [17].

### 2.3. Covariates

Household size was defined as the sum of all the AE within the household. Household composition was classified in four types: (1) Adults only, (2) adults and children only, (3) adults and adolescents only, and (4) adults, adolescents and children. Head of household educational level was classified according to the maximum degree of studies as: Low (from 0 to 6 years of study), medium (from 7 to 12 years of study) and high (≥13 years of study). Household women’s occupation was identified based on the type of employment that they reported and was classified as: Work inside home or outside home. Household income was deflated using November 2016 as a base and divided by the household AE; additionally, we stratified households in each survey into quintiles according to their income/AE. We only present quintiles I, III and V (very low, medium and very high). Place of residence was classified as rural (locations with <2500 inhabitants) and urban (locations with ≥2500 inhabitants). The country was divided into four regions with common sociodemographic characteristics: North region (Baja California, Baja California Sur, Chihuahua, Coahuila, Durango, Nuevo León, Sonora and Tamaulipas); Central region (Aguascalientes, Colima, Estado de México, Guanajuato, Jalisco, Michoacán, Morelos, Nayarit, Querétaro, San Luis Potosí, Sinaloa, and Zacatecas); Mexico City; and South region (Campeche, Chiapas, Guerrero, Hidalgo, Oaxaca, Puebla, Quintana Roo, Tabasco, Tlaxcala, Veracruz and Yucatán).

### 2.4. Data Analysis

We estimated means and proportions of sociodemographic characteristics to describe the households across the years.

We estimated the total expenditure and volume of purchases per day per AE, and for each NOVA food group the % consumers, % expenditure, and % volume in each survey. We performed linear regression models with a continuous variable for the years to estimate the linear trend (crude and adjusted by household size, household composition, head of household educational level, household women’s occupation, household income, residence area, and region). In this analysis, we present the linear regression coefficient (crude and adjusted) associated with a 10-year change.

We also estimated total energy purchased, and for each NOVA food group and subgroup the % energy purchased in each survey years. We performed the similar linear regression models previously described. In this analysis, we also present the linear regression coefficient (crude and adjusted) associated with a 10-year change.

Additionally, we estimated the mean of daily energy purchased (kcal/AE/day) of the NOVA food groups, and ultra-processed food subgroups across the years. For this estimation, to ease interpretation, we regrouped the 10 ultra-processed food subgroups into five: (1) Cookies, pastries, sweet bread, and breakfast cereals; (2) ultra-processed tortilla, bread and meats; (3) sugar-sweetened beverages (carbonated and non-carbonated sugar-sweetened beverages, and yoghurt and milk-based beverages); (4) sweet and salty snacks; and (5) other ultra-processed foods.

Finally, we assessed the trends in the % energy purchased of ultra-processed foods by household composition, head of household educational level, household women’s occupation, household socioeconomic status, residence area, and region. We performed multiple linear regression models (adjusted for all the covariates of the study) with an interaction term between the year (dummy variables) and each sociodemographic characteristic to estimate the adjusted mean of % energy purchased of ultra-processed foods for each year and sociodemographic characteristic strata. We assessed if the % energy purchased of ultra-processed foods in 2016, and the change from 1984 to 2016 had a statistically significant difference between sociodemographic characteristics.

All the analyses were performed using Stata version 12.0 (Stata Press, College Station, TX, USA) with the survey command to account for the design effect of complex surveys and weighting factors to obtain national representation. For some analyses, although they were conducted including all years, we only present the results of the ENIGH 1984, 1992, 2000, 2008 and 2016 to ease interpretation. We considered a *p* value < 0.05 to test statistical significance.

## 3. Results

### 3.1. Sociodemographic Characteristics Changes from 1984 to 2016

In Table 1, we present the sociodemographic characteristics in Mexican households from 1984 to 2016. Over time, the number of members within the household decreased from 4.5 to 3.3 AE, the proportion of households with adults and children decreased from 38.9% to 31.0%, the proportion of households headed by a person with high education increased from 7.1% to 16.3%, the proportion of women working outside home increased from 24.7% to 46.1%, the quarterly household income increased from 10,514.5 to 18,194.9 MXN/AE and the proportion of households living in urban areas increased from 76.9% to 78.4%.

### 3.2. Trends in Expenditure, Volume, and Percentage of Consumers of NOVA Food Groups

In Table 2, we present the trends in expenditure, volume, and % consumers of total purchases and by NOVA food groups in Mexican households from 1984 to 2016. Trends in all the indicators were statistically significant (*p* < 0.05). Total expenditure and volume of purchases decreased over time. The % consumers and the contribution to volume and expenditure of unprocessed or minimally processed foods, and processed culinary ingredients decreased over time; whereas processed food and ultra-processed foods increased. However, the magnitude of the change differed. Processed foods’ indicators showed the smallest change across the years. The largest change in % consumers was observed for processed culinary ingredients, with a reduction of 8.32 percentage points (pp) for every 10 years; followed by ultra-processed foods with an increase of 3.91 pp for every 10 years. On the other hand, the largest changes in the % expenditure and in the % volume were observed for ultra-processed foods (an increase of 3.80 and 4.58 pp for every 10 years, respectively). Moreover, both indicators of ultra-processed foods had the greatest relative changes, the % expenditure increased by 88.05% and the % volume increased by 125.70% between 1984 and 2016. When we adjusted these indicators for sociodemographic characteristics, we observed similar trends but with a smaller magnitude in general. For instance, the % expenditure in unprocessed or minimally processed foods changed from −3.08 to −2.07 pp for every 10 years; whereas in ultra-processed foods changed from 3.80 to 2.71 pp for every 10 years.

### 3.3. Trends of the Daily Total Energy Purchased and Energy Purchased Contribution of NOVA Food Groups

In Table 3, we present the trends of the daily total energy purchased and the contribution to total energy by each NOVA food group and subgroup in Mexican households from 1984 to 2016. For every 10 years (*p* < 0.05), the total energy purchased decreased 169.01 kcal/AE/day (from 2428.8 to 1875.4 kcal/AE/day from 1984 to 2016), the relative change was −22.78%. For NOVA food groups, for every 10 years, there was a decrease of 3.04 pp in % kcal of unprocessed or minimally processed foods (from 69.8% to 61.4% kcal from 1984 to 2016) and 1.79 pp in % kcal of processed culinary ingredients (from 14.0% to 9.0% kcal); while there was an increase of 0.42 pp in % kcal of processed foods (from 5.7% to 6.5% kcal) and 4.42 pp in % kcal of ultra-processed foods (from 10.5% to 23.1% kcal). Some of the subgroups with the largest absolute change (for every 10 years) in each NOVA food group were corn (−3.07 pp), oils and fats (−1.16 pp), cheeses (0.31 pp) and carbonated sugar-sweetened beverages (0.73 pp). Additionally, some of the food subgroups with the largest relative change (from 1984 to 2016), in each NOVA food group, were poultry and game (+147.83%), sweeteners (+34.00%), processed meat (+83.33%) and salty snacks (+900.00%). After adjustment for sociodemographic characteristics, we observed similar trends in NOVA food groups and subgroups but with a smaller magnitude; although for some of them the magnitude was larger.

### 3.4. Trends of the Daily Total Energy Purchased and Energy Purchased Contribution of NOVA Food Groups

Figure 1 shows the trends of the daily energy purchased (kcal/AE/day) of the NOVA food groups and ultra-processed foods subgroups in Mexican households from 1984 to 2016. For NOVA food groups, there was a decrease in unprocessed or minimally processed foods (from 1703.4 to 1126.0 kcal/AE/day) and processed culinary ingredients (from 378.2 to 218.7 kcal/AE/day); whereas, there was an increase in ultra-processed foods (from 225.8 to 414.9 kcal/AE/day). Meanwhile, processed foods remained relatively unchanged (from 121.5 to 115.8 kcal/AE/day). All the ultra-processed food subgroups increased across the years, among those with the largest change were cookies, pastries, sweet bread and breakfast cereals (from 117.5 to 153.3 kcal/AE/day); and ultra-processed tortilla, bread and meats (29.4 to 91.6 kcal/AE/day).

### 3.5. Trends of the Energy Purchased Contribution of Ultra-Processed Foods Stratified by Sociodemographic Characteristics

Figure 2 shows the trends in the daily energy purchased contribution of ultra-processed foods by sociodemographic characteristics from 1984 to 2016 adjusted by all other sociodemographic characteristics. We observed that the energy purchased contribution of ultra-processed foods in 2016 was higher (*p* < 0.05), and the change from 1984 to 2016 was larger (*p* < 0.05) in: Households in which the head of the family had a high educational level compared to those in which the head of the family had a low and medium educational level; households where the women work outside the home compared to those in where the women work inside the home; households with medium and very high socioeconomic status compared to those with low socioeconomic status; households from urban areas compared to those from rural areas; and households from the central and north region compared to those from the south region. On the other hand, there were no differences in the change from 1984 to 2016 among the households with different composition. However, we observed that the energy purchased contribution of ultra-processed foods was higher (*p* < 0.05) in 2016 in household with adults and adolescents, adults, adolescents and children, and adults and children compared to those with adults only.

## 4. Discussion

We studied the trends of food purchases over a one-week period from nationally representative samples of Mexican households. From 1984 to 2016, there was a decrease of unprocessed or minimally processed foods and processed culinary ingredients; however, in processed foods, there was a slight increase. On the other hand, for ultra-processed foods there was an important increase. The increase of ultra-processed foods was very sharp in earlier years, it increased from 10.5% kcal in 1984 to 22.3% kcal in 2006; whereas, from 2006 onwards, there were small increments and decrements with the highest point being 2012 with 23.7% kcal.

Ultra-processed foods are expanding in the food system across the globe. We found in Mexico, that the energy contribution of ultra-processed foods to household’s purchases increased from 10.5% kcal in 1984 to 23.1% kcal in 2016. Similar trends, but in higher magnitude, have been documented in previous studies of household food expenditure from Brazil (from 20.8% kcal in 2002 to 2003 to 25.4% kcal in 2008 to 2009) [18], Canada (from 24.4% kcal in 1938 to 1939, to 54.9% kcal in 2001) [19] and Spain (from 11.0% kcal in 1990 to 31.7% kcal in 2010) [20]. On the other hand, a wide range in the energy contribution of ultra-processed foods to household’s purchases has been found among several European countries: 13.4% kcal in Italy (1996), 36.9% kcal in Norway (1998), 46.2% kcal in Germany (1998), 26.4% kcal in Malta (2000), 13.7% kcal in Greece (2004), and 50.7% kcal in the UK (2008) [21]. Such differences in the energy contribution between countries could be due to different degrees of exposure to transnational food corporations’ retailing, manufacturing, and fast-food services over the last years, especially in high-income countries [22]. Therefore, while in high-income countries, such as Canada, the UK, and US, ultra-processed foods are more prominent [3,7,8]; in upper-middle income countries, such as Brazil and Mexico, unprocessed and minimally processed foods still coexist with ultra-processed foods in the diet [4,6].

Popkin et al. propose that income growth, policy liberalization, infrastructure improvement, urbanization and the rise of rural nonfarm employment have promoted changes in the food system and diet [23]. Moreover, a previous study in Mexico highlighted that income, urbanization, and region were the main drivers involved in dietary patterns changes over time, especially for unprocessed or minimally processed foods [24]. Hence, there are reasons to believe that some of these factors could be driving the trends of purchases of ultra-processed foods in Mexico over the last 32 years.

We observed that the household income in Mexico rose from $10,514.5 in 1984 to $18,194.9 MXN/AE/quarterly in 2016. Parallel to the increase in income, the energy contribution of ultra-processed foods to total purchases maintained, in general, an upward trend across the years; some of the few exceptions were in 1996 and 2008, where a slight decrease from the previous survey was observed. It is possible that these decreases were related to the economic crisis at the end of 1994 and mid-2008 [25].

At the beginning of the 1980s, Mexico faced a severe economic crisis characterized by the falling of international oil prices, the decline in the agricultural sector and the inability to pay international loans on time. To respond to this crisis, in 1986, Mexico joined the General Agreement on Tariffs and Trade (GATT) that facilitated the entrance of foreign food corporations [26]. Nevertheless, the trend of foreign direct investment of fast food restaurants and large food retailers increased when Mexico signed the North American Free Trade Agreement (NAFTA) in 1994. With this agreement, many fast food restaurant franchises (e.g., Kentucky Fried Chicken, Pizza Hut, Domino’s Pizza, McDonald’s) started to proliferate in Mexican cities [27]. On the other hand, the entrance of global supermarket chains in Mexico (e.g., Walmart, Costco, Sam’s Club, City Market) and local food retailers’ efforts to continue in the market increased the variety and availability of ultra-processed foods [28]. In our analysis, we observed that the energy contribution of ultra-processed foods increased rapidly between 1998 (15.8% kcal) and 2006 (22.4% kcal). However, in 2008, the energy contribution of ultra-processed foods to total purchases decreased to 21.4% kcal. The reason for this could be related to the global economic crisis that, in addition to the decrease in household’s income, led to a sharp decline in foreign food industry investment [25].

As urbanization increases, supermarkets, convenience stores, and fast-food restaurants expand and catch a greater number of consumers [22]. In parallel, the employment rate and working conditions play a role in consumers’ food choices [29]. In Mexico, urban areas have increased. With this demographic change, food chains establishments have also expanded. Overall, Mexico has a large net of approximately 5600 supermarkets, 17,500 convenience stores and 30,270 fast food chains restaurants [30,31]. Consequently, ultra-processed foods are becoming more accessible and available both in rural or urban localities [6]. Moreover, given that in Mexico the population spends more time at work [32] and that the participation of women in the labor market has increased (from 24.7% in 1984 to 46.1% in 2016), consumers might have less time to prepare healthy foods and therefore they demand more convenient ultra-processed food choices away-from-home [33].

Previous studies have found that ultra-processed foods consumption differs among strata of sociodemographic characteristics [3,5,6,8]. In our analysis, we observed that the highest and the largest increments in energy contribution of ultra-processed foods came from households: In which the head of the family has a high educational level; in which women work outside of the home; with a very high income; located in urban areas; and located in the North region. The most notorious differences among sociodemographic characteristics were found in household’s head educational level and household’s income. The difference between low and high education level, and low and high income was close to 10 pp; almost equivalent to the difference found between 2016 and 1984 in the whole population. Consistent with our results, in Chile and Brazil, ultra-processed foods consumption is positively associated with income [5,18]. On the other hand, in Canada and the US, the consumption of these products was relatively similar with income but negatively associated with educational level [3,8]. Further research is needed to understand how sociodemographic characteristics are associated with ultra-processed food across countries at different stages of the nutrition transition.

Interestingly, after we adjusted trends in the energy contribution of NOVA food groups and subgroups by sociodemographic characteristics (household size, household composition, household head educational level, household women’s occupation, income, residence area, and region) changes over time decreased in magnitude slightly but were still present in most food groups. This means that other factors, most likely environmental, macroeconomic or even cultural are also driving these changes. Moreover, there were some food subgroups such as corn tortilla, and milk, in which the adjusted trend was larger than the crude trend. This means that if sociodemographic variables, such as income and education, had not improved over time, the expenditure on corn tortilla should have increased more (because higher income and education are negatively associated with corn tortilla), and the expenditure on milk should have decreased more (because higher income and education are positively associated with milk).

We found that among unprocessed or minimally processed foods, from 1984 to 2016, there was an increase in the contribution of corn tortilla (from 20.4% to 23.3% kcal), and a decrease in corn (from 14.8% to 4.2% kcal). Although corn tortilla is unprocessed or minimally processed, it resembles ultra-processed foods because it is convenient and does not require any culinary preparation to be consumed. Whereas corn is used in traditional culinary preparations (e.g., pozole, tamales, tlacoyos, champurrado, atole). From a public health stand, it might be hard to limit the increased popularity of ready-to-eat products, but the creation and promotion of ready-to-eat products within the unprocessed or minimally processed foods could be an appealing alternative.

In our study, we found that the purchases of ultra-processed foods increased, not only in terms of energy contribution (from 10.5% kcal in 1984 to 23.1% kcal in 2016), but also in terms of absolute energy (from 225.8 to 414.9 kcal/AE/day). Moreover, this was the case even though the total kilocalories purchased decreased from 2428.8 to 1875.4 kcal/AE/day. The finding that total energy purchased decreased might be due to an increase in thepurchases away-from-home. Unfortunately, the type of food consumed away-from-home is not collected in this survey. Given the high number of food chain establishments in Mexico [30,31], future surveys need to include questions that can ascertain the type of foods consumed away-from-home.

In the last two surveys (2014 and 2016) we observed a decrease in the energy contribution of ultra-processed foods, especially from carbonated sugar-sweetened beverages and salty snacks. This could be due to the launch of the National Strategy for the Prevention and Control of Overweight, Obesity and Diabetes in 2014 [34], which included a 1 peso per liter tax to sugar-sweetened beverages (~10%), and an 8% tax to non-basic energy-dense foods. Previous studies agreed that the taxes achieved the reduction expected [35,36]. However, stronger measures that include improvements in the food system and the food environment, and behavior-change communication strategies are needed to reverse, more meaningfully, the upward trend of ultra-processed foods consumption experienced in the last decades.

Strengths of our study include the use of national representative and systematically collected information. This enables the analysis of a large time frame trend of food purchases in Mexico. On the other hand, by using the NOVA food framework, we could identify how ultra-processed foods are becoming part of the dietary patterns of Mexican households.

There are also limitations of our study. First, household’s food and beverage expenditure records do not consider waste and foods consumed away-from-home. Therefore, they do not accurately reflect dietary intake. For instance, in a previous analysis with 24-h recalls, from the 2012 National Health and Nutrition Survey, we found that the contribution of unprocessed or minimally processed foods, processed culinary ingredients, processed foods, and ultra-processed foods was 54.0%, 10.2%, 6.0%, and 29.8% kcal, respectively [6]; whereas in this study in 2012 the contribution was 60.9%, 9.1%, 6.2%, and 23.7% kcal, respectively. Hence, ultra-processed foods were underestimated, and unprocessed or minimally processed foods were overestimated with expenditure data. This might be due to a higher wastage of unprocessed or minimally processed foods, and a higher away-from-home intake of ultra-processed foods. Nevertheless, expenditure data are useful to assess long-term trends. Moreover, the associations found with sociodemographic variables in this study and in the previous 24-h recall analysis are comparable [6].

Another limitation of our analysis is that breakfast cereals were not collected from 1984 to 2000, affecting the estimation of the ultra-processed foods purchases’ trend. Next, processed foods and ultra-processed foods tend to be reformulated by food industries. Since in our study we used only one nutritional composition table across all surveys, it is possible that the energy purchased contribution of some of these products might be under-estimated or over-estimated in some time periods. Finally, since our study focused only on energy, further studies are needed to evaluate how the trend of ultra-processed foods is associated with the trends in macro and micronutrients.

## 5. Conclusions

Our study indicates that ultra-processed foods purchases in Mexico have doubled in the last three decades, paralleling a gradual decrease in unprocessed or minimally processed foods, and processed culinary ingredients purchases. Moreover, we found that this trend is only partially explained by sociodemographic changes in household size and composition, education level, income, women’s occupations, urbanization, and geographical region. Hence, other macroeconomic, environmental, and sociocultural factors must be important drivers of these trends. In order to improve the dietary patterns of Mexico, future food environment, food system, behavior-change communication and education strategies should promote the consumption of unprocessed or minimally processed foods such as fruits and vegetables, whole grains, nuts, and legumes; and discourage ultra-processed foods availability and accessibility.

## Figures and Tables

**Figure 1 nutrients-11-00045-f001:**
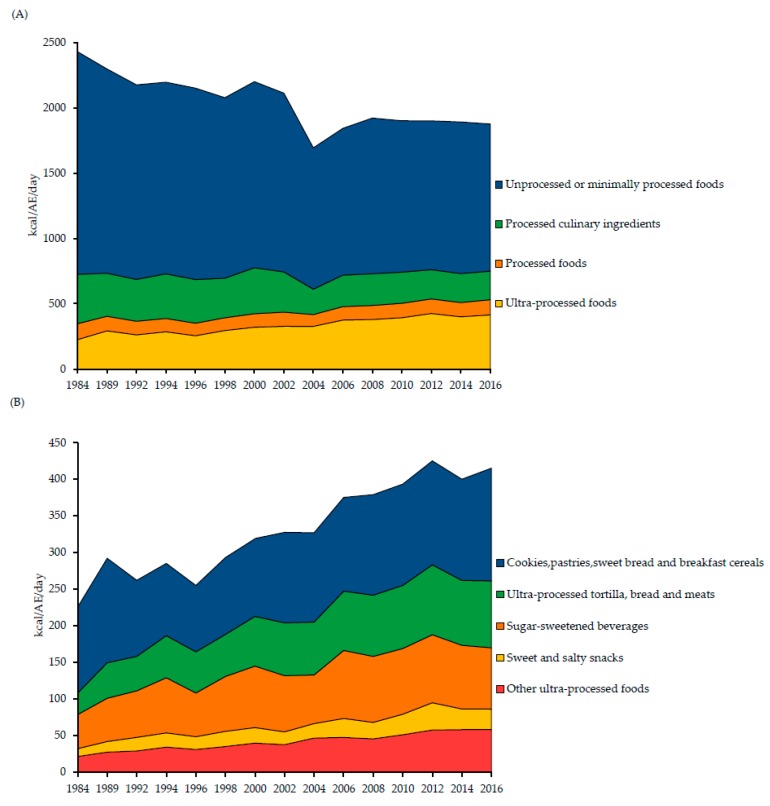
Trends over time in daily energy purchased (kcal/AE/day) of (**A**) NOVA food groups and (**B**) ultra-processed foods subgroups (ENIGH1984–2016).

**Figure 2 nutrients-11-00045-f002:**
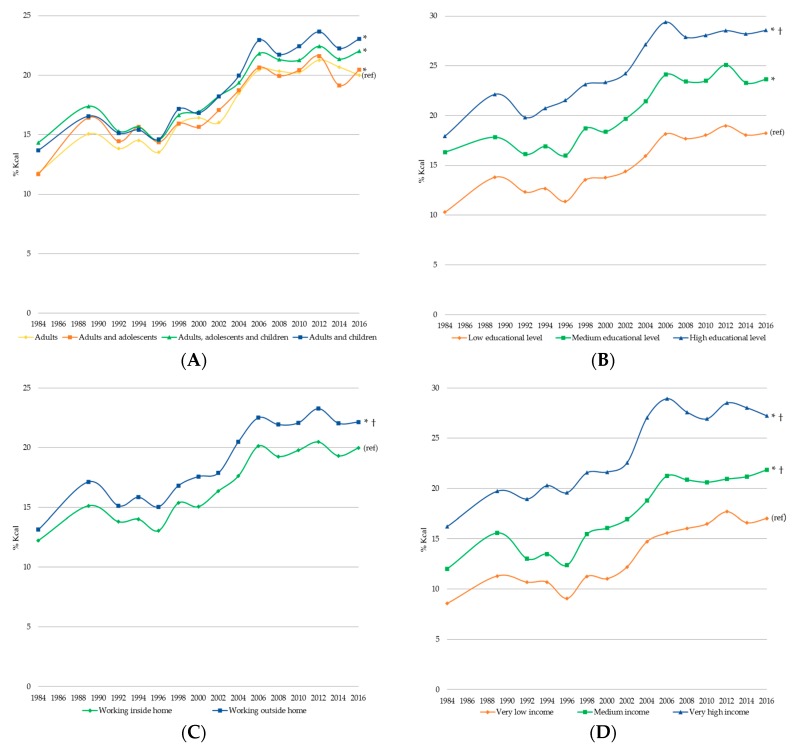
Trends in daily energy purchased contribution of ultra-processed foods stratified by (**A**) household composition, (**B**) head of household educational level, (**C**) household women’s occupation, (**D**) household income, (**E**) residence area and (**F**) region (ENIGH 1984–2016). Adjusted by household size and all other variables presented in the figure. * Value in 2016 was different (*p* < 0.05) vs. value in reference category. ^†^ Change from 1984 to 2016 was different (*p* < 0.05) vs. change in reference category.

**Table 1 nutrients-11-00045-t001:** Trends over time in demographic variables in Mexican households (National Income and Expenditure Survey (Encuesta Nacional de Ingresos y Gastos de los Hogares; ENIGH) 1984–2016).

	1984	1992	2000	2008	2016
Sample size, *n* households	4555	10,215	9565	28,244	67,807
Household size, AE ^a^	4.5	4.2	3.7	3.7	3.3
Household composition, %					
Adults only	17.9	20.3	27.2	31.1	37.0
Adults and children only	38.9	38.4	36.4	31.9	31.0
Adults and adolescents only	9.3	11.4	12.6	13.5	12.8
Adults, adolescents, and children	33.9	29.9	23.8	23.5	19.2
Head of household educational level, %					
Low education (0–6 years)	77.4	66.4	57.0	49.3	39.2
Medium education (7–12 years)	15.5	23.3	29.6	37.1	44.5
High education (≥13 years)	7.1	10.3	13.4	13.6	16.3
Women working outside the home, %	24.7	28.4	36.9	37.1	46.1
Household income ^b^, MXN ^c^/AE ^a^/quarterly	10,514.5	16,108.3	15,898.3	17,463.0	18,194.9
Household income ^b^ by quintiles, MXN ^c^/AE ^a^/quarterly					
Very low (quintile I)	2409.7	2561.8	2577.9	3454.2	4089.0
Medium (quintile III)	7128.6	7940.0	8343.9	10,737.5	10,903.5
Very high (quintile V)	28,225.9	42,606.1	42,947.8	50,884.5	47,385.2
Residence area, %					
Urban	76.9	76.6	77.9	78.8	78.4
Rural	23.1	23.4	22.1	21.2	21.6
Geographical region, %					
North	20.0	20.1	20.1	21.2	21.1
Central	32.6	29.3	29.6	32.1	29.3
Mexico City	23.1	20.4	19.8	16.1	18.9
South	24.3	30.2	30.5	30.6	30.7

^a^ Adult Equivalent. ^b^ Household income deflated by the national consumer price index (NCPI) of November 2016. ^c^ Mexican pesos.

**Table 2 nutrients-11-00045-t002:** Trends over time in daily total purchases and by NOVA food groups in Mexican households (ENIGH 1984–2016).

	1984	1992	2000	2008	2016	Linear Trend Coefficient (10-Year Increase)	Adjusted Linear Trend Coefficient ^a^ (10-Year Increase)	Relative Change (2016 vs. 1984)
**Total expenditure ^b^, MXN/AE/day**	30.6	30.2	25.8	25.9	27.3	−1.06 *	−2.79 *	−10.78%
**Total volume, g/AE/day**	1228.2	1195.9	1259.0	1162.6	1132.3	−28.58 *	−69.12 *	−7.81%
**Unprocessed or minimally processed foods ^c^**								
% consumers	99.5	99.5	99.0	98.9	98.9	−0.20 *	−0.19 *	−0.60%
% expenditure	72.8	71.4	66.9	64.6	64.0	−3.08 *	−2.07 *	−12.09%
% volume	78.7	75.8	71.8	68.0	67.3	−3.94 *	−2.95 *	−14.48%
**Processed culinary ingredients ^d^**								
% consumers	70.0	64.4	62.3	49.7	46.4	−8.32 *	−8.31 *	−33.71%
% expenditure	7.6	4.4	4.0	3.8	3.1	−0.97 *	−0.72 *	−59.21%
% volume	5.9	5.1	4.8	3.6	3.3	−0.88 *	−0.69 *	−44.06%
**Processed foods ^e^**								
% consumers	65.4	67.4	70.5	74.4	70.9	1.67 *	1.68 *	8.40%
% expenditure	6.3	7.0	7.3	7.2	7.7	0.26 *	0.08 *	22.22%
% volume	4.5	4.0	3.8	4.3	4.8	0.24 *	0.04 *	6.67%
**Ultra-processed foods ^f^**								
% consumers	82.9	85.5	92.3	95.6	95.0	3.91 *	3.91 *	14.60%
% expenditure	13.4	17.2	21.7	24.3	25.2	3.80 *	2.71 *	88.05%
% volume	10.9	15.1	19.5	24.1	24.6	4.58 *	3.60 *	125.70%

^a^ Adjusted by household size, household composition, head of household educational level, women’s occupation, household income, residence area and region. ^b^ Expenditure deflated by the national consumer price index (NCPI) of November 2016. ^c^ Includes corn tortillas; milk; red meat; cereals; fruits; poultry and game; beans; eggs; vegetables; corn; starchy vegetables; coffee and tea; and other unprocessed or minimally processed foods (dry peppers, lentils, chickpeas, nuts and almonds without salt and spices). ^d^ Includes oils and fats; sweeteners; and other processed culinary ingredients (salt, Mexican “mole” and vinegar). ^e^ Includes bread; cheeses; processed meat; and other processed foods (canned legumes, vegetables and fruits, salted nuts and seeds and Mexican cheeses). ^f^ Includes cookies, pastries and sweet bread; carbonated sugar-sweetened beverages; salty snacks; industrialized tortillas; and bread, candies and sweets; yoghurt and milk-based beverages; sausages and other ultra-processed meats; non-carbonated sugar sweetened beverages; and other ultra-processed foods (baby food products, ice cream, pizza, ultra-processed cheeses and dressings). * *p* < 0.05 for linear coefficient to test if there was any significant change for every 10 years.

**Table 3 nutrients-11-00045-t003:** Trends over time in total daily energy purchased and energy contribution of NOVA food groups and subgroups in Mexican households (ENIGH 1984–2016).

	1984	1989	1992	1994	1996	1998	2000	2002	2004	2006	2008	2010	2012	2014	2016	Linear Trend Coefficient (10-Year Increase)	Adjusted Linear Trend Coefficient ^a^ (10-Year Increase)	Relative Change (2016 vs. 1984)
**Total energy, kcal/AE ^b^/day**	**2428.8**	**2296.4**	**2174.9**	**2195.0**	**2149.7**	**2076.6**	**2199.5**	**2111.4**	**1692.9**	**1841.9**	**1920.8**	**1900.0**	**1899.3**	**1890.9**	**1875.4**	**−169.01 ***	**−196.22 ***	**−22.78%**
NOVA groups and subgroups, % kcal																		
**Unprocessed or minimally processed foods**	**69.8**	**67.7**	**69.0**	**67.7**	**68.5**	**66.8**	**65.2**	**65.4**	**67.1**	**61.6**	**62.7**	**62.3**	**60.9**	**62.3**	**61.4**	**−3.04 ***	**−1.89 ***	**−12.03%**
Corn tortillas	20.4	19.3	22.0	22.4	24.1	23.6	22.3	23.6	28.6	23.3	23.1	23.4	22.3	23.4	23.3	0.71 *	1.30 *	14.22%
Milk	7.2	6.7	6.9	6.7	6.3	6.8	6.4	6.7	6.7	5.8	5.9	5.7	5.2	4.9	4.8	−0.78 *	−1.14 *	−33.33%
Red meat	4.9	5.6	5.9	5.8	5.1	5.8	6.0	4.8	3.5	4.0	3.9	4.0	3.9	3.4	3.6	−0.83 *	−0.96 *	−26.53%
Cereals (except corn)	4.9	5.0	5.0	4.8	5.0	5.2	5.1	4.5	4.4	4.3	4.8	4.9	4.8	5.1	4.6	−0.08 *	0.12 *	−6.12%
Fruits	2.0	2.0	2.3	2.3	2.1	2.0	2.5	2.6	2.4	2.5	2.3	2.3	2.7	2.6	2.5	0.17 *	−0.04	25.00%
Poultry and game	2.3	2.9	4.0	4.2	3.6	3.8	4.3	4.2	4.2	5.4	5.1	4.9	5.0	5.1	5.7	0.87 *	0.75 *	147.83%
Beans	7.1	6.4	6.2	5.9	6.3	5.7	5.2	5.0	3.9	4.4	4.6	4.6	4.2	4.6	3.9	−0.92 *	−0.55 *	−45.07%
Eggs	3.2	3.2	3.3	3.2	3.3	3.4	3.4	3.3	3.3	3.8	4.0	4.3	3.6	4.0	4.3	0.36 *	0.35 *	34.38%
Vegetables	1.2	1.5	1.5	1.4	1.6	1.5	1.6	1.8	1.6	1.8	1.9	2.0	1.9	2.1	2.1	0.27 *	0.25 *	75.00%
Corn	14.8	13.2	10.0	9.2	9.3	7.3	6.7	7.1	6.6	4.2	4.9	4.1	4.8	4.7	4.2	−3.07 *	−2.23 *	−71.62%
Starchy vegetables	1.0	1.0	1.0	0.9	1.0	0.9	1.0	0.9	0.9	0.9	1.0	1.0	1.1	1.1	1.1	0.05 *	0.07 *	10.00%
Coffee and tea	0.2	0.2	0.2	0.2	0.2	0.2	0.2	0.2	0.2	0.2	0.2	0.3	0.3	0.2	0.3	0.05 *	0.03 *	50.00%
Seafood	0.2	0.3	0.2	0.2	0.2	0.2	0.2	0.2	0.4	0.4	0.3	0.3	0.3	0.3	0.3	0.05 *	0.04 *	50.00%
Other ^c^	0.5	0.5	0.5	0.5	0.4	0.4	0.4	0.5	0.6	0.7	0.6	0.6	0.7	0.8	0.7	0.10 *	0.12 *	40.00%
**Processed culinary ingredients**	**14.0**	**12.8**	**12.5**	**13.2**	**13.3**	**12.1**	**13.3**	**11.9**	**7.9**	**10.1**	**10.0**	**10.0**	**9.1**	**9.3**	**9.0**	**−1.79 ***	**−1.38 ***	**−35.71%**
Oils and fats	9.0	8.1	7.8	8.6	8.7	7.7	8.8	7.5	4.9	6.3	6.5	6.5	5.9	5.8	5.6	−1.16 *	−0.88 *	−37.78%
Sweeteners	5.0	4.7	4.6	4.6	4.6	4.3	4.4	4.3	3.0	3.8	3.4	3.4	3.1	3.4	3.3	−0.64 *	−0.51 *	−34.00%
Other ^d^	0.0	0.0	0.0	0.0	0.0	0.0	0.0	0.0	0.0	0.1	0.1	0.1	0.1	0.0	0.1	0.01 *	0.01 *	-
**Processed foods**	**5.7**	**5.6**	**5.4**	**5.1**	**4.9**	**5.3**	**5.2**	**5.5**	**5.4**	**6.0**	**6.0**	**6.0**	**6.2**	**6.1**	**6.5**	**0.42 ***	**0.20 ***	**14.04%**
Bread	3.8	2.8	2.6	2.2	2.0	2.0	1.7	1.9	2.3	2.1	2.4	2.2	2.3	2.1	2.2	−0.20 *	−0.19 *	−42.11%
Cheeses	0.8	1.2	1.2	1.2	1.2	1.4	1.4	1.5	1.4	1.7	1.6	1.7	1.7	1.9	1.9	0.31 *	0.25 *	137.50%
Processed meat	0.6	0.6	0.6	0.6	0.7	0.7	0.7	1.4	0.8	0.9	0.9	0.9	1.1	1.0	1.1	0.18 *	0.17 *	83.33%
Other ^e^	0.5	1.0	1.0	1.0	1.0	1.2	1.4	0.8	1.0	1.3	1.0	1.1	1.2	1.1	1.3	0.12 *	-0.03	160.00%
**Ultra-processed foods**	**10.5**	**13.9**	**13.2**	**14.0**	**13.2**	**15.8**	**16.3**	**17.2**	**19.5**	**22.3**	**21.4**	**21.7**	**23.7**	**22.3**	**23.1**	**4.42 ***	**3.06 ***	**120.00%**
Cookies, pastries and sweet bread	5.1	6.5	5.0	4.6	4.5	5.4	5.2	5.7	6.3	6.2	6.4	6.2	6.7	6.4	6.8	0.62 *	0.31 *	33.33%
Carbonated sugar-sweetened beverages	1.7	2.1	2.7	2.9	2.6	3.5	3.6	3.6	3.8	4.7	4.3	4.2	4.2	3.9	4.0	0.73 *	0.48 *	135.29%
Salty snacks	0.1	0.2	0.1	0.3	0.3	0.4	0.4	0.4	0.6	0.8	0.7	0.8	1.2	0.9	1.0	0.35 *	0.30 *	900.00%
Industrialized tortillas and bread	0.7	1.1	1.2	1.3	1.3	1.3	1.6	1.9	2.2	2.5	2.3	2.4	2.5	2.3	2.7	0.64 *	0.43 *	285.71%
Candies and sweets	0.3	0.5	0.6	0.5	0.5	0.6	0.6	0.5	0.6	0.6	0.5	0.6	0.7	0.6	0.5	0.05 *	0.01	66.67%
Yoghurt and milk-based beverages	0.5	0.6	0.7	0.9	0.7	0.8	0.9	0.8	0.8	1.0	0.9	0.9	1.0	1.0	0.9	0.12 *	0.04 *	80.00%
Sausages and other ultra-processed meats	0.8	1.3	1.3	1.6	1.7	1.7	1.9	1.9	2.0	2.4	2.5	2.5	2.8	2.7	2.6	0.60 *	0.52 *	225.00%
Non-carbonated sugar sweetened beverages	0.2	0.3	0.3	0.3	0.2	0.3	0.3	0.1	0.1	0.7	0.6	0.6	0.6	0.6	0.6	0.17 *	0.13 *	200.00%
Breakfast cereals ^f^	-	-	-	-	-	-	-	0.5	0.6	0.8	0.7	0.8	0.8	0.8	0.8	0.37 *	0.31 *	-
Other ^g^	1.1	1.3	1.3	1.5	1.5	1.7	1.8	1.8	2.6	2.6	2.5	2.7	3.1	3.1	3.2	0.76 *	0.54 *	190.91%

^a^ Adjusted by household size, household composition, head of household educational level, women’s occupation, household income, residence area and region. ^b^ Adult Equivalent. ^c^ Includes nuts and seed (unsalted); other legumes; dried herbs. ^d^ Includes chicken and beef broth; Mexican “mole”; condiments. ^e^ Includes nuts and seeds (salted); salted, dried or oil-preserved canned fish and meat; canned fruits; vegetables; legumes; beer, cider and wine. ^f^ For the food subgroup breakfast cereals, there was any food item to be grouped in ENIGH 1984–2000. ^g^ Includes baby food; pizza; hot-dog; hamburger; dressings; whisky, rum and vodka. * *p* < 0.05 for linear coefficient to test if there was any significant change for every 10 years. The bold is necessary to notice the principal food groups.

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
