# Peer review of "Trends in Ultra-Processed Food Purchases from 1984 to 2016 in Mexican Households"

_nutrients, 2018, doi:10.3390/nu11010045_

Round 1
Reviewer 1 Report
A final line-by-line proof reading is needed to catch final typographical/grammar/compositional errors including but not limited to:
Line 24 Change ‘double’ to ‘doubled’
Lines 83 and other lines throughout using ‘e.g.,’. Remove the ‘etc’ at the end of the listings, as this is redundant.
Line 153. Comma after ‘In Table 1,’ and Line 190 after same prepositional phrase starting the sentence.
Line 189. Proof read all table headings to ensure consistent capitalization according to journal style. Shouldn’t the first instance of ‘purchased’ also be capitalized (here and in other tables)?
After line 221, Panel A. This graphic needs to be reset so that the vertical title has comparable distance from the units as in Panel B. Similarly, the legend boxes to the right of the graph in Panel A should be similarly distanced as in Panel B. Ensure these are parallel in construction.
Line 264. Shouldn’t this be “corporations’ “
Line 284 Edit to ‘To respond to this crisis,…’
Line 288 Check official spelling (capitalization) of Pizza Hut.
Line 304 Edit to ‘in the labor’
Line 305. Last part of sentence needs editing between ‘…consumers……. …[33].’
Line 308. I believe the singular is stratum and the plural is ‘strata.
Lines 329 Edit ‘would not had improved’ to ‘had not improved’
Lines 330 and 331 Edit ‘would had’ here and in the next line to ‘should have’
Line 341. Shouldn’t ‘purchased of’ be ‘contained in’?
Line 359. Edit to ‘Strengths of our study,’ and Line 363 edit to ‘…limitations of our study,’
Line 372. Technically, it should be ‘data are’
Line 376. I believe the study would benefit from a final sentence about nutrition (not simply energy, as is the focus of the study). I would suggest:
Line 376 Change ‘Finally, to ‘Next,’
Then a last sentence be included something like:
‘Finally, our study focused on energy composition of various groups of foods. Further studies are needed to examine how the consumption of high energy foods from whatever source impacts nutrition including marco and micronutrients. ‘
Line 385. Edit to ‘women’s occupations’
Line 386 Change to ‘drivers of these…’
Line 387. I suggest editing to ‘..preserve the more healthful dietary patterns,..’
Author Response
RESPONSE TO REVIEWER 1
A final line-by-line proof reading is needed to catch final typographical/grammar/compositional errors including but not limited to:
Response: Thank you for your comment, we did a line-by-line proof reading in addition to revising the specific comments below.
Line 24 Change ‘double’ to ‘doubled’
Response: Done in line 25.
Lines 83 and other lines throughout using ‘e.g.,’. Remove the ‘etc’ at the end of the listings, as this is redundant.
Response: Thank you very much; we removed ‘etc’ in every sentence that use ‘e.g.’
Line 153. Comma after ‘In Table 1,’ and Line 190 after same prepositional phrase starting the sentence.
Response: Done in line 160 and line 199. We put comma after ‘In Table 1’, ‘In Table 2’ and ‘In Table 3’
Line 189. Proof read all table headings to ensure consistent capitalization according to journal style. Shouldn’t the first instance of ‘purchased’ also be capitalized (here and in other tables)?
Response: Done. Yes they should be capitalized.
After line 221, Panel A. This graphic needs to be reset so that the vertical title has comparable distance from the units as in Panel B. Similarly, the legend boxes to the right of the graph in Panel A should be similarly distanced as in Panel B. Ensure these are parallel in construction.
Response: Done. Panel A of Figure 1 was modified according to your specifications.
Line 264. Shouldn’t this be “corporations’ “
Response: Done in line 277.
Line 284 Edit to ‘To respond to this crisis,…’
Response: Done in line 297.
Line 288 Check official spelling (capitalization) of Pizza Hut.
Response: Done in line 301.
Line 304 Edit to ‘in the labor’
Response: Done in line 318.
Line 305. Last part of sentence needs editing between ‘…consumers……. …[33].’
Response: Agree. From line 319 to 321, we rewrite this sentence as follows: consumers might have no time for preparing healthy foods and hence they demand for more convenient ultra-processed food choices found away-from-home.
Line 308. I believe the singular is stratum and the plural is ‘strata.
Response: Done in line 323.
Lines 329 Edit ‘would not had improved’ to ‘had not improved’
Response: Done in line 343.
Lines 330 and 331 Edit ‘would had’ here and in the next line to ‘should have’
Response: Done in line 344 and 346.
Line 341. Shouldn’t ‘purchased of’ be ‘contained in’?
Response: ’Contained in’ might be inexact but we revised this sentence to improve clarity in line 356.
Line 359. Edit to ‘Strengths of our study,’ and Line 363 edit to ‘…limitations of our study,’
Response: Done in line 375 and 379, respectively.
Line 372. Technically, it should be ‘data are’
Response: Agree. We made this change in line 388.
Line 376. I believe the study would benefit from a final sentence about nutrition (not simply energy, as is the focus of the study). I would suggest:
Line 376 Change ‘Finally, to ‘Next,’
Response: Done in line 392.
Then a last sentence be included something like:
‘Finally, our study focused on energy composition of various groups of foods. Further studies are needed to examine how the consumption of high energy foods from whatever source impacts nutrition including marco and micronutrients. ‘
Response: Agree. We include this last sentence “Finally, our study focused only on energy. Further studies are needed to evaluate how the trends of ultra-processed foods is associated with the trends in macro and micronutrients.”
Line 385. Edit to ‘women’s occupations’
Response: Done in line 404.
Line 386 Change to ‘drivers of these…’
Response: Done in line 405.
Line 387. I suggest editing to ‘..preserve the more healthful dietary patterns,..’
Response: We revised this sentence to improve clarity in line 406.

Reviewer 2 Report
This is a very well written MS on a very interesting research topic.
The authors have addressed sufficiently all the underlying parameters. Their conclusions are valid.
Given, though, the increasing trends in BMI worldwide, some suggestions could be added in the conclusions part. These suggestions should be towards the application of Mediterranean diet globally.
As reported here
https://www.cambridge.org/core/journals/public-health-nutrition/article/changing-the-irish-dietary-guidelines-to-incorporate-the-principles-of-the-mediterranean-diet-proposing-the-medeire-diet/5D8E9328111EE18583916ADF2104499A
In Ireland the national food pyramid has mistakes and a better food pyramid should be proposed. The improved pyramid should include more information for the Med diet pyramid. Similar actions could be taken all over the world since Med diet has been shown that it is the most healthy dietary pattern around the World.
I would suggest minor revision of the MS.
Happy to review the revised MS, if needed.
Author Response
RESPONSE TO REVIEWER 2
This is a very well written MS on a very interesting research topic.
The authors have addressed sufficiently all the underlying parameters. Their conclusions are valid.
Given, though, the increasing trends in BMI worldwide, some suggestions could be added in the conclusions part. These suggestions should be towards the application of Mediterranean diet globally.
As reported here
https://www.cambridge.org/core/journals/public-health-nutrition/article/changing-the-irish-dietary-guidelines-to-incorporate-the-principles-of-the-mediterranean-diet-proposing-the-medeire-diet/5D8E9328111EE18583916ADF2104499A
In Ireland the national food pyramid has mistakes and a better food pyramid should be proposed. The improved pyramid should include more information for the Med diet pyramid. Similar actions could be taken all over the world since Med diet has been shown that it is the most healthy dietary pattern around the World.
Response: Thank you very much for your comment and the reference provided. We agree that we should add more information on the types of foods and dietary patterns that should be promoted within the unprocessed or minimally processed foods to achieve a healthy dietary pattern. Hence we added that the intake of fruits and vegetables, whole grains, nuts, and legumes should be promoted. These food groups are the core of the Mediterranean diet and also take into account the Mexican context, and are in line with the Mexican dietary guidelines. We would rather not recommend explicitly the adoption of the Mediterranean diet in Mexico, because for example a key component of it, the olive oil, is very expensive in Mexico and not used in traditional culinary dishes.
I woulld suggest minor revision of the MS.
Happy to review the revised MS, if needed.

Reviewer 3 Report
The topic of the study is important and really interesting. In my opinion, the aim, results, discussion and the research limitations are formulated in a clear way.
However, I would recommend to explain the NOVA food framework in the abstract of the manuscript in order to make the ‘NOVA name’ more understandable for readers who are not familiar with this kind if classification.
Moreover, I would also recommend (in conclusions) to indicate that education of the consumers will be one of the options of encouraging them to consume the unprocessed or minimally processed foods.
Author Response
RESPONSE TO REVIEWER 3
The topic of the study is important and really interesting. In my opinion, the aim, results, discussion and the research limitations are formulated in a clear way.
However, I would recommend to explain the NOVA food framework in the abstract of the manuscript in order to make the ‘NOVA name’ more understandable for readers who are not familiar with this kind if classification.
Response: Agree. In the abstract, we described that we used the NOVA food framework to classify foods according to their degree of processing and added the four NOVA food groups of this framework: 1) unprocessed or minimally processed foods; 2) processed culinary ingredients; 3) processed foods; and 4) ultra-processed foods.
Moreover, I would also recommend (in conclusions) to indicate that education of the consumers will be one of the options of encouraging them to consume the unprocessed or minimally processed foods.
Response: Agree. Education is key for encouraging unprocessed or minimally processed foods (UNMP). We were implicitly referring to education in our suggestion of using “behavior-change communication” strategies; however, we added education to make it explicit.
